# Incremental Pose Map Optimization for Monocular Vision SLAM Based on Similarity Transformation

**DOI:** 10.3390/s19224945

**Published:** 2019-11-13

**Authors:** Wenlei Liu, Sentang Wu, Zhongbo Wu, Xiaolong Wu

**Affiliations:** 1School of Automation Science and Electrical Engineering, Beihang University, Beijing 100191, China; woost@sina.com; 2Institute of Automation, Chinese Academy of Sciences, Beijing 100190, China; wuzhongbo17@mails.ucas.ac.cn; 3Navigation and Control Technology Institute of NORINCO Group, Beijing 100089, China; wuaero@buaa.edu.cn

**Keywords:** similarity transformation, incremental pose map, monocular vision SLAM, bag-of-words, sparse direct method, histogram equalization, probability graph

## Abstract

The novel contribution of this paper is to propose an incremental pose map optimization for monocular vision simultaneous localization and mapping (SLAM) based on similarity transformation, which can effectively solve the scale drift problem of SLAM for monocular vision and eliminate the cumulative error by global optimization. With the method of mixed inverse depth estimation based on a probability graph, the problem of the uncertainty of depth estimation is effectively solved and the robustness of depth estimation is improved. Firstly, this paper proposes a method combining the sparse direct method based on histogram equalization and the feature point method for front-end processing, and the mixed inverse depth estimation method based on a probability graph is used to estimate the depth information. Then, a bag-of-words model based on the mean initialization K-means is proposed for closed-loop feature detection. Finally, the incremental pose map optimization method based on similarity transformation is proposed to process the back end to optimize the pose and depth information of the camera. When the closed loop is detected, global optimization is carried out to effectively eliminate the cumulative error of the system. In this paper, indoor and outdoor environmental experiments are carried out using open data sets, such as TUM and KITTI, which fully proves the effectiveness of this method. Closed-loop detection experiments using hand-held cameras verify the importance of closed-loop detection. This method can effectively solve the scale drift problem of monocular vision SLAM and has strong robustness.

## 1. Introduction

With the continuous development of artificial intelligence technology, research of visual slam has also made great progress, and new achievements are emerging. This technology plays an important role in vision navigation. Vision slam is widely used in robot autonomous navigation, augmented reality, 3D reconstruction, unmanned driving, and other fields. According to the different equipment used, visual SLAM can be divided into visual SLAM based on a depth camera RGB-D, visual SLAM based on a binocular or structural camera, and visual SLAM based on a monocular camera. The difficulty of the above three methods gradually increases, and each method has its own advantages and disadvantages [1,2]. For an RGB-D camera, it enjoys the advantages of obtaining the depth information of the object more accurately, which is conducive to the accuracy of SLAM positioning; but it is limited by a narrow field of vision, limited measurement range, poor anti-interference capacity, and is mostly used indoors. The binocular camera has a strong anti-interference capacity and depth information can be obtained utilizing its baseline. In addition, the binocular camera can be used indoors and outdoors. However, the camera configuration and calibration of the binocular camera are complex, and the positioning accuracy is restricted by the baseline of the binocular camera. Moreover, with a large amount of calculation, hardware equipment is usually required for acceleration. When it comes to the monocular camera, it is superior due to its simple structure, low cost, and wide range of use. The disadvantage is that the monocular camera cannot obtain the depth information of objects, which can only be obtained by the depth estimation method. Additionally, the monocular camera has the problem of scale drift, so its accuracy is relatively poor. This paper focuses on the problem of monocular visual SLAM. Aiming at the uncertainty of depth measurement by the monocular camera, an incremental pose map optimization method based on similarity transformation is proposed by using the inverse depth estimation method based on a probability graph for scale drift.

The optical flow method and direct method are widely used in visual SLAM. Usually, the calculated camera pose is used to provide a good initial value for further optimization. The optical flow method was introduced in detail, and some improvement measures were proposed for the traditional optical flow method, such as describing most of the algorithms and their extensions in a consistent framework, and an effective inverse synthesis algorithm is proposed in [3]. Engel et al. [4] gave a detailed introduction to the sparse direct method, which combines the minimization of photometric errors with the uniform joint optimization of all model parameters, independent of key points and descriptors. In [5,6], aiming at the low computational speed and lack of consistency of the direct method, a semi-direct method was proposed. This method eliminates the feature extraction and robust matching techniques of motion estimation, and has the characteristics of high accuracy, good robustness, and fast speed. In [7], a monocular vision inertial ranging algorithm based on the Extended Kalman Filter (EKF) method and the direct photometric error minimization method was proposed. The data of the vision and inertial measurement were fused. Aiming at the problems of the optical flow method and direct method in practical application, a sparse direct method based on histogram equalization was proposed, which can effectively guarantee the photometric invariance in the measurement process.

Depth estimation plays an important role in monocular vision SLAM. The accuracy of depth estimation will directly affect the location accuracy of monocular vision SLAM. Civera et al. [8] applied the inverse depth parametrization estimation to monocular vision SLAM, which can measure features in a large depth range and reduce the influence of parallax on depth estimation. However, directly parameterizing inverse depth increases the complexity of parameter optimization. Engel et al. [9] presented a novel real-time range measurement method by a monocular camera. The key of this method is to continuously estimate the half-density inverse depth map of the current frame and use the image alignment to track the camera motion. Perdices et al. [10] used semi-direct visual localization (SDVL) to estimate the inverse depth, and used the exception rejection mechanism to eliminate the dislocation features, thus improving the efficiency and accuracy of feature matching. In [11], a method of depth estimation based on Bayesian estimation and convex optimization image processing was proposed. Newcombe et al. [12] proposed a real-time density tracking and mapping method, which estimates detailed texture depth mapping on selected key frames to generate surface mosaics composed of vertices. Inverse depth estimation is widely used in visual odometry because of its high accuracy and good robustness. In this paper, a mixed inverse depth estimation method based on probability graph is proposed.

The estimation and optimization of the pose of the monocular camera is the core content of visual SLAM. There are many methods for pose optimization. The more accurate the pose estimation is, the higher the measurement accuracy is. Triggs et al. [13] gave a detailed introduction to the bundle adjustment methods commonly used in camera measurement, including the selection of the cost function, and the introduction of numerical optimization methods, like the sparse Newton method and the linear approximation recursive method. The re-projection was adopted to eliminate errors, which greatly improved the accuracy of estimation. Strasdat et al. [14] proposed a two-level optimization method, which combines the precise point constraints of the main regions of interest with the stable edges of the position-attitude soft constraints, dynamically choosing the inner and outer window members and optimizing them simultaneously. Dubbelman et al. [15] proposed a closed-loop online pose chain method to solve the pose, which can accurately estimate the depth of visual odometry and reduce the influence of scale drift on pose estimation. In [16], an open-square smoothing filtering algorithm was proposed to estimate the pose of the camera. This method decomposes the correlation information matrix or the measured Jacobian matrix into square roots, so the calculation speed is faster, and the accuracy is higher. In [17], a new data structure, namely the Bayesian tree algorithm, was introduced. The method of a probability graph was used to infer the pose, and sparse matrix decomposition was combined so as to improve the speed and accuracy of operation. Kaess et al. [18] adopted incremental smoothing and mapping to decompose the information matrix, in which only the matrix items that actually changed were updated, thus improving the efficiency. In addition, the uncertainty estimation algorithm based on the factor information matrix was utilized to improve the real-time information processing. In order to reduce the amount of calculation and ensure the accuracy of the optimization, the incremental pose map optimization method is adopted in this paper.

The commonly used loop closure detection method for monocular vision SLAM is bag-of-words, which establishes a dictionary by extracting the features of all images, and then searches for matches through the dictionary. In addition to this, other methods have been proposed by researchers. Redmond et al. [19] presented a method of initializing the k-means clustering algorithm, which uses the kd-tree to estimate the density of data at different locations, and then uses a modification of the Katsavounidis algorithm. Arthur et al. [20] proposed a simple random seeding technique to increase k-means, and obtained a Θ(logk)-competition and optimal clustering algorithm. In [21], a probabilistic recognition method based on position representation was proposed, which can not only be used for positioning, but also determine that a new observation comes from a new place to enhance its map. Cummins et al. [22] proposed a new SLAM formula based on the appearance only, which is suitable for very large-scale local recognition. The system navigates in outer space and assigns each new observation value to a new or previously visited location without reference to the metric location. Che et al. [23] proposed an image retrieval framework based on robust distance measurement and information fusion, which improved the retrieval performance of SVT relative to the baseline. In [24], a new method based on the stacked denoise autoencoder (SDA) was proposed. SDA is a multi-layer neural network, which learns compression representation from original input data independently in an unsupervised way. In order to prevent the dictionary from falling into the local optimum in the process of dictionary generation, a K-means clustering method based on mean initialization is proposed in this paper.

According to the different application environments and conditions of visual SLAM, scholars have proposed various solutions. In [25], a new direct tracking method based on *sim*(3) was proposed to detect scale drift explicitly and incorporate the effect of the noisy depth value into the tracking. The algorithm allows the construction of large-scale and consistent environmental maps. Mur-Artal et al. [26,27] presented a feature-based monocular SLAM system, namely Oriented FAST and Rotated BRIEF SLAM (ORB-SLAM). The system has strong robustness to severe motion clutter, allows wide baseline loops to be closed and repositioned, and includes automatic initialization. It has a faster operation speed and higher accuracy. Zhou et al. [28] proposed a semi-dense monocular simultaneous location and mapping (SLAM) method that can deal with pure camera rotation motion, and established a probabilistic depth map model based on Bayesian estimation. Frost et al. [29] proposed a novel monocular vision SLAM method. In addition to point measurement, target detection is also considered to solve the scale ambiguity and drift. By expecting the size of the object, the scale estimation can be seamlessly integrated into a bundle adjustment. In [30], the relative position of the first key frame was calculated by using the calibrated object, and the depth map was initialized. Meanwhile, the metric was introduced into the reconstruction, so that the absolute scale of the world coordinate system could be obtained. In order to eliminate the influence of scale drift on positioning accuracy, an incremental pose map optimization method based on similarity transformation is proposed in this paper.

The system architecture of this paper is shown in Figure 1. It consists of three threads: Front-end processing, closed-loop detection, and back-end processing. Different processing methods are provided in different threads.

In front-end processing, this paper combined the direct method with the feature point method, using the direct method in a certain threshold range, and the feature point method when it exceeded a certain threshold. In the direct method, the equalization based on a histogram is proposed, which can ensure that the assumption of gray invariance is satisfied in the process of the direct method. In this paper, a mixed inverse depth estimation method based on a probability graph was used to estimate the depth information of monocular vision, which not only improves the accuracy but also improves the robustness. In the closed-loop detection, this paper used the appearance-based bag-of-words. In the process of dictionary generation, the K-means clustering method based on mean initialization was proposed to ensure the global optimal characteristics of cluster centers. The data structure was analyzed by the Thiessen polygon, which is convenient for dictionary representation, and a similar scoring function is expressed by information theory. In the back-end processing, the main innovation is to propose a method of incremental pose optimization based on similarity transformation, which takes the uncertainty of scale into account to ensure the consistency of monocular vision slam. At the same time, the incremental pose optimization facilitates the expansion of observation nodes and reduces the computational complexity. Reference frames and key frames are determined according to the number of feature points, and global optimization is carried out by using the results of closed-loop detection. In addition, the detected closed-loop features can be used for global optimization to further reduce the error.

The organizational structure of this paper is as follows: Section 2 will focus on the front-end processing method based on the combination of the direct method and feature point method; Section 3 will focus on the closed-loop detection based on bag-of-words; Section 4 will mainly introduce the method of incremental pose map optimization based on similarity transformation; Section 5 will introduce indoor environment experiments, outdoor environment experiments, and closed-loop detection experiments of hand-held cameras; and Section 6 will summarize the methods provided in this paper.

## 2. Front-End Processing

In visual SLAM, the front end is called visual odometry (VO). According to the information of adjacent images, the motion of the camera is roughly estimated, and the initial value is provided for back-end processing. Common VO can be divided into two categories:

(a) Direct method: The direct method can be used to estimate the motion of the camera directly by using the gray information of the image, without calculating the key points and descriptors, as long as there are light and shade changes in the scene. The direct method uses the gradient size and direction of the local gray intensity to optimize, which gets rid of the dependence on image features. In the case of the camera motion being blurred, it has strong robustness. The direct method evolved from the optical flow method. The optical flow method describes the motion in the pixel, while the direct rule takes into account the motion model of the camera at the same time. The assumption of gray invariance must be taken into account when using direct method; that is, the grayscale value of a pixel in the same spatial point should remain unchanged in each image.

(b) Method based on feature points: The usual processing methods include the following: Firstly, distinct features, such as feature points or feature lines, are extracted from each image; secondly, invariant feature descriptors are used to match them; and then, epipolar geometry is used to restore the pose of the camera; finally, the pose is optimized by minimizing the re-projection error. The classical invariant feature extraction method is ORB feature extraction, which mainly consists of key points and descriptors. Through Features From Accelerated Segment Test (FAST) corner extraction, the obvious changes of local pixels can be obtained, and the main direction of feature points can be calculated at the same time. The Binary Robust Independent Elementary Features (BRIEF) descriptor is used to describe the image region around the extracted feature points, which is suitable for real-time image matching. The advantage of the feature point method is that it runs stably, is insensitive to light and dynamic objects, and has strong robustness. The disadvantage is that the extraction of feature points and the calculation of descriptors are time-consuming, and the processing error of images without obvious texture features is relatively large.

The feature point method is suitable for scenes with abundant feature points, and is easy to extract. Because enough feature points contain more redundant information, the calculation of the pose will be more accurate. The direct method gets rid of the dependence on feature points, and calculates the pose of the camera by using the gray changes of the front and back frames. It can be applied to scenes with fewer feature points. In this paper, the direct method and the feature point method are combined to make full use of the advantages of both, so as to make full use of the advantages and avoid disadvantages. It not only extends the application scope and scene of the vision front-end but also improves the accuracy of front-end processing.

The results of front-end processing will be taken as the initial value of back-end optimization. The accuracy of the calculation will affect the results of the back-end optimization, and the speed of the calculation will affect the real-time performance of the whole SLAM. In order to ensure the accuracy of pose estimation and the speed of operation, a relatively simple switching method is adopted, which uses the number of feature points and gray changes to judge whether to use the feature point method or the direct method. When the number of feature points is less than a certain threshold and the gray level changes of the two frames are not obvious, the sparse direct method based on a histogram equalization is adopted; otherwise, front-end processing based on the ORB feature is used. According to the different application scenarios, the threshold can be adjusted appropriately to get better optimization results.

### 2.1. Sparse Direct Method Based on Histogram Equalization

In order to satisfy the hypothesis of an invariant gray level, histogram equalization is usually used to preprocess the image to eliminate the influence of strong and weak light on the gray level characteristics. The histogram is a statistical method that uses graphics to represent the distribution of data. Usually, the abscissa represents the grayscale value corresponding to the pixel, and the ordinate represents the number of statistical samples per pixel.

Histogram equalization is an image enhancement method that enhances image contrast by stretching the range of image pixels. For an image, the important feature pixels are usually concentrated in a certain gray interval. Therefore, stretching the uniform distribution of the gray interval in the whole gray level range can make the contrast of the features significantly enhanced and the image has a strong sense of clarity. Histogram equalization reduces the area with high grayscale value in the image, increases the area with low grayscale value, and compresses the gray level with a lesser number of pixels, expanding the gray interval with a greater number of pixels. The method of histogram equalization has the advantages of fast operation, better reflection of the details of the image, effective restraint of the drastic changes of brightness between images, reduction of the difference of the grayscale values between adjacent images, and better maintenance of the gray consistency of images.

In the sparse direct method, the estimation of camera pose can be computed with fewer feature points without calculating the descriptor or matching feature. The basic idea is to estimate the position of corresponding matching points by using the current camera pose. By minimizing the photometric error, the pose of the camera is optimized. The optimal error function is:(1)$$\underset{\xi }{\min}\Delta \left(\xi \right)=\sum_{i=1}^{N}{\left\|e_{i}\right\|}^{2}=\sum_{i=1}^{N}{\left\|I\left(p_{1,i}\right)-I\left(p_{2,i}\right)\right\|}^{2}$$
where $I\left(p_{1,i}\right)$ is the grayscale value of the pixel of the *i*-th feature point in the first image and $I\left(p_{2,i}\right)$ is the grayscale value of the pixel in the second image corresponding to the *i*-th feature point in the first image. *N* is the number of feature points.

In the second image, the corresponding feature point, *p*_2*,i*_, is obtained by the camera pose transformation. *R* is the rotation matrix; *t* is the translation vector; ξ is the Lie algebra of *R* and *t*; $\rho_{i}$ is the corresponding inverse depth information; and *K* is the camera’s internal parameter matrix. The conversion formula is as follows:(2)$$p_{2,i}=\rho_{i}K\left(RP_{i}+t\right)=\rho_{i}K\left(\exp\left(\xi^{\wedge }\right)P_{i}\right)$$

In this paper, the perturbation model of Lie algebra is used to solve the error. After Lie algebra left-multiplies by a small perturbation, δξ, we can get:(3)$$\begin{array}{ll} e\left(\xi ⊕\delta \xi \right) & =I_{1}\left(\rho_{1}KP\right)-I_{2}\left(\rho_{2}K\exp\left(\delta \xi^{\wedge }\right)\exp\left(\xi^{\wedge }\right)P\right)\\ & \approx I_{1}\left(\rho_{1}KP\right)-I_{2}\left(\rho_{2}K\exp\left(\xi^{\wedge }\right)P+\rho_{2}K\delta \xi^{\wedge }\exp\left(\xi^{\wedge }\right)P\right) \end{array}$$

The coordinate of the perturbation component in the second camera coordinate system is set as *q*, namely, $q=\delta \xi^{\wedge }\exp\left(\xi^{\wedge }\right)P$, and its corresponding pixel coordinate is *m*, namely, $m=\rho_{2}Kq=\rho_{2}K\delta \xi^{\wedge }\exp\left(\xi^{\wedge }\right)P$. By using the first order Taylor formula, the error function can be transformed into the following formula:(4)$$\begin{array}{ll} e\left(\xi ⊕\delta \xi \right) & \approx I_{1}\left(\rho_{1}KP\right)-I_{2}\left(\rho_{2}K\exp\left(\xi^{\wedge }\right)P\right)-\frac{\partial I_{2}}{\partial m}\frac{\partial m}{\partial q}\frac{\partial q}{\partial \xi }\delta \xi \\ & =e\left(\xi \right)-\frac{\partial I_{2}}{\partial m}\frac{\partial m}{\partial q}\frac{\partial q}{\partial \xi }\delta \xi =e\left(\xi \right)-J\left(\xi \right)\delta \xi \end{array}$$
where $\frac{\partial I_{2}}{\partial m}$ is the gradient of pixels at *m*. $\frac{\partial m}{\partial q}$ is the derivative of the pixel coordinates in the camera coordinate system, and $\frac{\partial q}{\partial \xi }$ is the derivative of the spatial point to Lie algebra. By calculating the point $q={\left[X,Y,Z\right]}^{T},Z=1/\rho$ in the camera coordinate system, the following formula can be obtained:(5)$$\frac{\partial m}{\partial \xi }=\frac{\partial m}{\partial q}\frac{\partial q}{\partial \xi }=\left[\begin{array}{c} \begin{array}{ccc} \rho f_{x} & 0 & \begin{array}{ccc} -\rho^{2}f_{x}X & -\rho^{2}f_{x}XY & \begin{array}{cc} f_{x}+\rho^{2}f_{x}X^{2} & -\rho f_{x}Y \end{array} \end{array} \end{array}\\ \begin{array}{ccc} 0 & \rho f_{y} & \begin{array}{ccc} -\rho^{2}f_{y}Y & -f_{y}-\rho^{2}f_{y}Y^{2} & \begin{array}{cc} \rho^{2}f_{y}XY & \rho f_{y}X \end{array} \end{array} \end{array} \end{array}\right]$$

Therefore, we can find that the Jacobian matrix of Lie algebra is $J=-\frac{\partial I_{2}}{\partial m}\frac{\partial m}{\partial \xi }$. Using the LM algorithm to solve the error function, the optimized pose can be obtained.

### 2.2. Front-End Processing Based on ORB Features

Based on the front-end processing of ORB features, the ORB features of each image are extracted first, then the fast library for approximate nearest neighbor (FLANN) algorithm is used to match all features, and finally the position and pose are optimized by the bundle adjustment (BA)algorithm using matched feature points.

Assuming that the coordinates of a point in three-dimensional space are $U_{i}={\left[X_{i},Y_{i},Z_{i}\right]}^{T}$, the coordinates transformed into camera coordinates are $U^{\prime }={\left[X^{\prime },Y^{\prime },Z^{\prime }\right]}^{T},\rho^{\prime }=1/Z^{\prime }$, and the projected pixel coordinates are $\mu_{i}={\left[u_{i},v_{i}\right]}^{T}$. The inner parameter matrix of the camera is *K*, and the Lie algebras of the camera’s pose, *R* and *t*, are recorded as ξ. The transformation relationship between the spatial point and pixel coordinate system is as follows:(6)$$s_{i}\mu_{i}=K\exp\left(\xi^{\wedge }\right)U_{i}=KU^{\prime }=\left[\begin{array}{ccc} f_{x} & 0 & c_{x}\\ 0 & f_{y} & c_{y}\\ 0 & 0 & 1 \end{array}\right]\left[\begin{array}{c} X^{\prime }\\ Y^{\prime }\\ Z^{\prime } \end{array}\right]$$

After eliminating the proportional coefficient, *s_i_*, the error function can be obtained as follows:(7)$$\{\begin{array}{c} e_{u}=f_{x}\frac{X^{\prime }}{Z^{\prime }}+c_{x}-u_{i}=\rho^{\prime }f_{x}X^{\prime }+c_{x}-u_{i}\\ e_{v}=f_{y}\frac{Y^{\prime }}{Z^{\prime }}+c_{y}-v_{i}=\rho^{\prime }f_{y}Y^{\prime }+c_{y}-v_{i} \end{array}$$

The Jacobian matrix, *J_p_*, of the re-projection error can be obtained by deriving the pose using the chain rule:(8)$$J_{p}=\frac{\partial e}{\partial \delta \xi }=\frac{\partial e}{\partial U^{\prime }}\frac{\partial U^{\prime }}{\partial \delta \xi }$$

The first is the derivative of the error with respect to the projection point. It can be obtained that:(9)$$\frac{\partial e}{\partial U^{\prime }}=\left[\begin{array}{c} \begin{array}{ccc} \frac{\partial e_{u}}{\partial X^{\prime }} & \frac{\partial e_{u}}{\partial Y^{\prime }} & \frac{\partial e_{u}}{\partial Z^{\prime }} \end{array}\\ \begin{array}{ccc} \frac{\partial e_{v}}{\partial X^{\prime }} & \frac{\partial e_{v}}{\partial Y^{\prime }} & \frac{\partial e_{v}}{\partial Z^{\prime }} \end{array} \end{array}\right]=\left[\begin{array}{c} \begin{array}{ccc} \frac{f_{x}}{Z^{\prime }} & 0 & -\frac{f_{x}X^{\prime }}{{Z^{\prime }}^{2}} \end{array}\\ \begin{array}{ccc} 0 & \frac{f_{y}}{Z^{\prime }} & -\frac{f_{y}Y^{\prime }}{{Z^{\prime }}^{2}} \end{array} \end{array}\right]$$

The second term is the derivative of the transformed point on Lie algebra. It can be obtained that:(10)$$\frac{\partial U^{\prime }}{\partial \delta \xi }=\left[\begin{array}{cc} I & {U^{\prime }}^{\wedge } \end{array}\right]=\left[\begin{array}{cc} \begin{array}{ccc} 1 & 0 & 0\\ 0 & 1 & 0\\ 0 & 0 & 1 \end{array} & \begin{array}{ccc} 0 & -Z^{\prime } & Y^{\prime }\\ Z^{\prime } & 0 & -X^{\prime }\\ -Y^{\prime } & X^{\prime } & 0 \end{array} \end{array}\right]$$

Therefore, the Jacobian matrix of the re-projection error can be obtained as follows:(11)$$J_{p}=\frac{\partial e}{\partial \delta \xi }=\left[\begin{array}{cc} \begin{array}{c} \begin{array}{ccc} \rho^{\prime }f_{x} & 0 & -{\rho^{\prime }}^{2}f_{x}X^{\prime } \end{array}\\ \begin{array}{ccc} 0 & \rho^{\prime }f_{y} & -{\rho^{\prime }}^{2}f_{y}Y^{\prime } \end{array} \end{array} & \begin{array}{c} \begin{array}{ccc} {\rho^{\prime }}^{2}f_{x}X^{\prime }Y^{\prime } & -f_{x}-{\rho^{\prime }}^{2}f_{x}{X^{\prime }}^{2} & \rho^{\prime }f_{x}Y^{\prime } \end{array}\\ \begin{array}{ccc} f_{y}+{\rho^{\prime }}^{2}f_{y}{Y^{\prime }}^{2} & -{\rho^{\prime }}^{2}f_{y}X^{\prime }Y^{\prime } & -\rho^{\prime }f_{y}X^{\prime } \end{array} \end{array} \end{array}\right]$$

The $\underset{\xi }{\min}\Delta \left(\xi \right)=\sum_{i=1}^{N}{\left\|e_{i}\right\|}^{2}=\sum_{i=1}^{N}{\left\|h\left(x_{i}\right)-\mu_{i}\right\|}^{2}$ is solved by using the method of non-linear least squares, and the optimal pose is finally obtained.

### 2.3. Mixed Inverse Depth Estimation Based on the Probability Graph

In the process of initialization, depth estimation is very important to the SLAM operation results of monocular vision. However, the accuracy and robustness of depth estimation are poor and vulnerable to external noise. In [31,32], in order to improve the accuracy and robustness, a mixed inverse depth estimation method based on a probability map is adopted.

Figure 2 is the probability diagram of the Gauss-uniform mixture probability distribution model [32]. Assume that $X=\left\{x_{1},\cdots ,x_{N}\right\}$ is the inverse depth value of the sensor observation, $\rho =\left\{\rho_{1},\cdots ,\rho_{N}\right\}$ is the real inverse depth value, $\pi =\left\{\pi_{1},\cdots ,\pi_{N}\right\}$ is the proportion of good measurement data, 1-*π* is the proportion of interference, and the interference signal obeys the uniform distribution form, $U[\rho_{\min},\rho_{\max}]$. $\rho_{\min},\rho_{\max}$ are the minimum inverse depth and the maximum inverse depth measured by the sensor, respectively. $\lambda =\left\{\lambda_{1},\cdots ,\lambda_{N}\right\}$ is the accuracy of the Gauss distribution, where λ=1τ, in which τ is the variance of the Gauss distribution. Given the true inverse depth, ρ, the accuracy, λ, of the Gauss distribution, and the proportional coefficient, π, of the correct data, the probability distribution of the inverse depth measured by the mixed model is as follows:(12)$$p(x_{n}|\rho_{n},\lambda_{n},\pi_{n})=\pi_{n}N(x_{n}|\rho_{n},\lambda_{n}{}^{-1})+(1-\pi_{n})U(x_{n})$$

All potential discrete variables are recorded as $Z=\{z_{1k},z_{2k},\cdots ,z_{nk}\}$, in which *z_ik_* is a binary random variable. Using the expression of “1-of-K”, one element is 1, the rest are 0. Among them, z_i1_ = 1, which means the *i*th measurement value is good measurement data, and z_i0_ = 0 which means the ith measurement value is interference data. Therefore, the distribution of potential variables is as follows:(13)$$p(x_{n}|\rho_{n},\lambda_{n},\pi_{n},z_{nk})={\left(\pi_{n}N(x_{n}|\rho_{n},\lambda_{n}{}^{-1})\right)}^{z_{nk}}{\left((1-\pi_{n})U(x_{n})\right)}^{1-z_{nk}}$$

In this paper, the conjugate prior probability distribution of parameter ρ,λ,π is introduced, in which the conjugate prior probability of the mixing coefficient, π, obeys a Beta distribution, and the conjugate prior distribution of the mean, ρ, and precision, λ, is a Gauss-Gamma distribution.

According to the Bayesian theorem, posterior∝likelihood×prior, the joint probability distribution of all random variables can be obtained in the form of:(14)P(X,Z,π,ρ,λ)=p(X|Z,ρ,λ)p(Z|π)p(U|Z,π)p(ρ|λ)p(λ)p(π)

By using the method of variational inference to estimate the parameters, the recursive formula of inverse depth estimation can be obtained.

## 3. Closed-Loop Detection

With the sequence movement of the camera, the camera pose and the depth information estimated by depth estimation will produce errors. Periodic global back-end optimization for the whole SLAM system will eliminate some errors, but there will still be some accumulated errors. In order to eliminate cumulative errors, a closed-loop detection method is usually used, which eliminates the cumulative errors in a larger time scale through similar data collected by the camera in the same place, so that the whole SLAM has a globally consistent estimation.

Appearance-based bag-of-words (BoW) is a common method for the detection of a closed loop. Bag-of-words usually uses the features of an image to describe an image. A word is used to represent a feature, and all the features are composed into a dictionary. Therefore, in the matching process, only the word vectors contained in each image need to be matched. The bag-of-words improves the efficiency and accuracy of closed-loop detection.

### 3.1. Dictionary Generation

Dictionary generation is a clustering problem and an unsupervised machine learning method. The learning classification is automatically performed according to the features extracted from the image, which improves the efficiency of dictionary generation. The common clustering method is the K-means clustering method, which is simple and effective. In this paper, a K-means clustering method based on mean initialization is proposed, so that the selection of clustering centers is not trapped in the local optimum, so as to ensure its global optimum characteristics.

Assuming that D is the feature point data set and C is the cluster center set, the K-means clustering method based on mean initialization is [33]:

(1) Center point initialization: The coordinates of all feature points are summed up, and then the average is taken. The feature point closest to the average is taken as the first cluster center, c_1_. If many points are identical, the point near the edge is selected as the first cluster center, and c_1_ is added to set C. Then, the feature point furthest from c_1_ is selected as the second cluster center, c_2_, which is also added to set C. Then, other centers are chosen according to the following formula. Among them, $\underset{c_{j}\in C}{\min}\left(d(x_{i},c_{j})\right)$ means to select the center point nearest to the feature point, calculate its Euclidean distance, and then select the feature point corresponding to the largest distance among all the distances as the new cluster center, and add it to set C:(15)$$\underset{x_{i}\in D}{\max}\left(\underset{c_{j}\in C}{\min}\left(d(x_{i},c_{j})\right)\right)$$

(2) For each feature point, the distance between each feature point and each center point is calculated and the point is classified into the nearest center points.

(3) According to the existing classification, the average value of feature points in each class is used to recalculate the central point, *c_j_*, of each class, and then the sum of squares, *D*, of the new cumulative distance is calculated, where n is the number of all feature points:(16)$$D={\sum_{i=1}^{n}\left(\underset{c_{j}\in C}{\min}\left(d(x_{i},c_{j})\right)\right)}^{2}$$

(4) If the change of *D* is small and convergent, then the algorithm converges, and the selected center point is suitable. Otherwise, the calculation will be restarted, returning to the second step.

Through the K-means clustering algorithm, feature points can be divided into K specific classes, so as to generate K words; that is, K words represent an image. The words of all images can form a dictionary. The dictionary is sorted according to the nature of the words and a vector is generated. This is not only convenient for the generation of dictionaries but also for the word searching. According to the words provided by the dictionary, feature matching can be carried out efficiently.

### 3.2. Layered Thiessen Polygon Data Structure

In order to facilitate the establishment and search of dictionaries, a K-ary tree is usually used to express a dictionary, while the K-ary tree can be abstractly represented by a Thiessen polygon [34], and its hierarchical way is shown in the following figure.

Figure 3 shows the generation process of the K-ary tree dictionary by using the Thiessen polygon. The classification using the Thiessen polygon is mainly based on the fact that the distance from any feature point in a Thiessen polygon to the feature control point constituting the polygon is less than the distance to the feature control point of other polygons. The feature control point here is the cluster center of the K-means clustering method. This paper takes the ternary tree as an example to introduce its principle. Among that, the green point is the control point. According to the main control point of each layer, a multi-layer Thiessen polygon can be generated. The classification results of each layer use the K-means clustering algorithm initialized by the mean. In the figure, A_1_, A_2_, and A_3_ are the results of the first layer classification; A_11_, A_12_, and A_13_ are the results of the classification of A_1_; and A_121_, A_122_, and A_123_ are the results of the classification of A_12_. Among them, the whole image is equivalent to the root node of the K-ary tree, the number of classifications in each layer is equivalent to the branch of the K-ary tree, and the number of layers is the depth of the K-ary tree. According to the above method, the whole image can be divided into n regions, which are the leaf nodes of the K-ary tree. All the leaf nodes make up n words of the whole image.

The words generated by the layered Thiessen polygon are convenient for the dictionary generation and word searching of the dictionary, which ensures the searching efficiency of the dictionary.

### 3.3. Computation of the Similarity Scoring Function

Assuming that the number of occurrences of the word *W_i_* in an image is *n_i_* and the image contains *n* words, the probability of the occurrence of the word *W_i_* in an image is P(Wi)=nin. According to information theory, it can be concluded in this paper that in the same image, each word contains a different amount of information. The words with a higher occurrence frequency often contain less information, while the words with a lower probability contain more information. The logarithmic function of inverse probability is usually used to express the amount of information contained in a word with the expression of log(1P(Wi))=−log(P(Wi))=log(nni). In information theory, the degree of uncertainty of a word measured in an image is expressed by self-entropy as follows [35]:(17)$$g\left(W_{i}\right)=P\left(W_{i}\right)\log\left(\frac{1}{P\left(W_{i}\right)}\right)=-P\left(W_{i}\right)\log\left(P\left(W_{i}\right)\right)=\frac{n_{i}}{n}\log\left(\frac{n}{n_{i}}\right)$$

Therefore, in an image, the uncertainty vector of the word *W_i_* composed by all words in the dictionary is as follows:(18)$$G={\left[g\left(W_{1}\right),g\left(W_{2}\right),\cdots ,g\left(W_{m}\right)\right]}^{T}$$
where *m* is the number of all words contained in the dictionary.

Supposing that *G_a_* is used to represent the uncertainty vector of image *a* and *G_b_* is used to represent that of image *b*, the similarity between the two images can be calculated with the similarity score function. The expression is as follows:(19)$$Y=1-{\left\|\frac{G_{a}}{{\left\|G_{a}\right\|}_{2}}-\frac{G_{b}}{{\left\|G_{b}\right\|}_{2}}\right\|}_{1}=1-\sum_{i=1}^{m}\left|\frac{g_{a}\left(W_{i}\right)}{{\left\|G_{a}\right\|}_{2}}-\frac{g_{b}\left(W_{i}\right)}{{\left\|G_{b}\right\|}_{2}}\right|$$

When the value of the similarity score reaches a certain threshold, it can be considered that image *a* is similar to image *b* and a closed loop is constituted. With the closed-loop characteristic, the accumulative error caused by the camera in the sequence motion process can be eliminated, and the positioning accuracy of visual SLAM can be greatly improved.

## 4. Back-End Processing

The front-end processing calculates the pose of the moving object, and its error is relatively large. Usually, this calculation result provides the initial value for the back-end optimization of the pose, and selects the key frame from the sequence image for optimization. The initial value of the scale factor is provided by the inverse depth estimation method based on a probability graph. The back end of this paper uses the method based on a similar transformation to optimize, and considers the role of the scale factor. In short, from the front end to the back end is a rough-to-fine continuous optimization process. The front-end processes the data preliminarily, and the back end processes the data more precisely, so that the positioning accuracy can be continuously improved while ensuring the real-time performance.

In monocular vision SLAM, one of the most important problems is scale uncertainty and scale drift. This paper presented an incremental pose map optimization method based on the similarity transformation group, which effectively solves the problems of rotation, translation, and scale drift in closed-loop detection.

Similarity transformation is a combination of Euler transformation and uniform transformation. Generally, the matrix of similar transformation is expressed as:(20)Sim(3)={S=[sRt01]∈ℝ4×4},s∈ℝ+(3),R∈SO(3),t∈ℝ3
where *s* is the scale scaling factor; *R* is the 3 × 3 rotation matrix; and *t* is the 3 × 1 translation vector. There is a total of seven degrees of freedom. The invariants of similar transformation are the ratio of two lengths and the included angle between two straight lines.

*sim*(3), the corresponding Lie algebra of *Sim*(3) is:(21)sim(3)={ς|ς=[ρϕσ]∈ℝ7,ς∧=[σI+ϕ∧ρ00]∈ℝ4×4}
where the element of Lie algebra *sim*(3) is a seven-dimensional vector, ς; ρ corresponds to the translation part; ϕ corresponds to the rotation part; and σ corresponds to the scale scaling factor. In the derivation process of this paper, the relevant formulas about similarity transformation are shown in Appendix A.

Figure 4 is a schematic diagram of an incremental pose map optimization method based on similarity transformation, in which each node represents the relevant random quantity. Among them, *x_i_* represents the node of the camera pose; *l_i_* is the landmark node; *o_ij_* is the observation node of the *j*-th landmark by the *i*-th camera; *f_ij_* is the pose error factor function, representing the pose relationship error value of the *j*-th camera relative to the *i*-th camera; and *q_i_* is the re-projection error factor function, representing the re-projection error of the *i*-th camera.

As can be seen from Figure 4, when the camera moves to the new position, *x_n_*_+1_, the pose of *x_n_*_+1_ will be optimized by the re-projection error factor function, *q_n_*_+1_. The pose factor function, *f_n(n_*_+1*)*_, is used to correlate with the original pose, *x_n_*, and the relative pose variation is determined to obtain the motion information of the camera. When the camera moves to a new position, the above process will be repeated. Therefore, the formation of an incremental motion calculation process only requires calculation of the motion relationship between the new and the original pose. It does not need to optimize the global information every time, which avoids a lot of repeated calculation, reduces the calculation cost, and improves the calculation speed.

The selection of key frames and reference frames has an important impact on the results of incremental pose map optimization. In this paper, the following principles are usually applied to the selection of key frames: 1. After global optimization, 20 frames are passed. 2. At least 100 feature points can be extracted from the image. 3. The number of matching points of the current frame is reduced by 80% compared with the reference frame. In general, key frames with less than 20% matching points of the reference frames are regarded as new reference frames.

In order to improve the computing speed and reduce the data storage, redundant key frames need to be removed. The method is to delete the remaining two key frames when the matching degree of feature points of the key frames of three consecutive images exceeds 90% and the middle frame is retained, which can reduce the number of key frames. At the same time, it reduces unnecessary closed loops and improves the efficiency of closed-loop detection.

Assuming that the pose error factor function is a zero-mean Gauss distribution:(22)$$f_{ij}\left(x_{i},x_{j}\right)=\frac{1}{\sqrt{2\pi *\Sigma_{ij}{}^{2}}}\exp\left(\frac{{\left(\ln\left(S_{ij}{}^{-1}S_{i}{}^{-1}S_{j}\right)\right)}^{2}}{2\Sigma_{ij}{}^{2}}\right)$$
where $S_{ij}=S_{i}{}^{-1}S_{j}$, *S_ij_* is the similarity transformation of pose *x_j_* relative to pose *x_i_*; *S_i_* is the similarity transformation of pose *x_i_* relative to the original pose *x*_0_; and *S_j_* is the similarity transformation of pose *x_j_* relative to the original pose *x*_0_. $\Sigma_{ij}$ is the covariance matrix of the pose error. Usually, in order to simplify the calculation, it can be made as a unit matrix.

Supposing that the re-projection error factor function satisfies the following form of a Gaussian distribution:(23)$$q_{i}\left(x_{i}\right)=N\left(h\left(x_{i}\right)|\mu_{i},\sigma_{i}{}^{2}\right)=\frac{1}{\sqrt{2\pi *\sigma_{i}{}^{2}}}\exp\left(\frac{{\left(h\left(x_{i}\right)-\mu_{i}\right)}^{2}}{2\sigma_{i}{}^{2}}\right)$$
where $h\left(x_{i}\right)$ is the re-projection of the *i*-th camera; *μ*_i_ is the corresponding pixel coordinate; $h\left(x_{i}\right)-\mu_{i}$ is the re-projection error; and $\sigma_{i}{}^{2}$ is the variance of the re-projection.

The joint probability distribution of all variables is expressed in the form of a factor product:(24)$$F\left(X\right)=\frac{1}{Z}\prod_{i}q_{i}\left(x_{i}\right)\prod_{i,j}f_{ij}\left(x_{i},x_{j}\right)$$
where $Z=\sum_{i,j}\prod_{i}q_{i}\left(x_{i}\right)\prod_{i,j}f_{ij}\left(x_{i},x_{j}\right)$, *Z* is usually called the normalization factor to ensure that F(X) is a properly defined probability. It is difficult to accurately calculate the value of *Z*, and in practice, it is not necessary to obtain the exact value of *Z*.

The maximum posteriori probability inference takes the form of:(25)$$X_{\max}=\underset{X}{\arg\max}F\left(X\right)=\underset{X}{\arg\max}\frac{1}{Z}\prod_{i}q_{i}\left(x_{i}\right)\prod_{i,j}f_{ij}\left(x_{i},x_{j}\right)$$

After taking logarithm of the upper formula, ignoring the normalization factor and other constant terms, the maximum posteriori probability inference problem can be transformed into the problem of a nonlinear least squares sum:(26)$$X_{\max}=\underset{X}{\arg\min}\frac{1}{2}\left({\sum_{i}\left\|h\left(x_{i}\right)-\mu_{i}\right\|}^{2}+\sum_{i,j}{\left(\ln\left(S_{ij}{}^{-1}S_{i}{}^{-1}S_{j}\right)\right)}^{2}\right)$$

There is an important defect in the formula. If there is a mismatch in the calculation, and the error is very large, it will dominate the whole error, erase the influence of other correct errors, and produce wrong results. In order to solve the problem of the large error and fast growth of two norms, this paper adopted the Huber kernels method. The improved formula is as follows:(27)$$\begin{array}{l} X_{\max}=\underset{X}{\arg\min}\left(\sum_{i}H\left(h\left(x_{i}\right)-\mu_{i}\right)+\sum_{i,j}H\left(\ln\left(S_{ij}{}^{-1}S_{i}{}^{-1}S_{j}\right)\right)\right),\\ H\left(h\left(x_{i}\right)-\mu_{i}\right)=\{\begin{array}{c} \frac{1}{2}{\left(h\left(x_{i}\right)-\mu_{i}\right)}^{2},\left|h\left(x_{i}\right)-\mu_{i}\right|\leq \beta_{1},\\ \beta_{1}\left(\left|h\left(x_{i}\right)-\mu_{i}\right|-\frac{1}{2}\beta_{1}\right),other. \end{array}\\ H\left(\ln\left(S_{ij}{}^{-1}S_{i}{}^{-1}S_{j}\right)\right)=\{\begin{array}{c} \frac{1}{2}{\left(\ln\left(S_{ij}{}^{-1}S_{i}{}^{-1}S_{j}\right)\right)}^{2},\ln\left(S_{ij}{}^{-1}S_{i}{}^{-1}S_{j}\right)\leq \beta_{2},\\ \beta_{2}\left(\ln\left(S_{ij}{}^{-1}S_{i}{}^{-1}S_{j}\right)-\frac{1}{2}\beta_{2}\right),other. \end{array} \end{array}$$
where $\beta_{1}$ is the threshold of the re-projection error and $\beta_{2}$ is the threshold of the pose error. The main function of the Huber kernel function is that when the error exceeds the threshold, it changes from a quadratic function to a primary function, which limits the growth rate of the error and weakens the influence of mismatching on the optimization results.

Firstly, the Jacobian matrix, *J_p_*, of the re-projection error function is solved by the BA method, and the re-projection function is linearized to obtain:(28)$$h\left(x_{i}\right)=h\left(x_{i}{}^{0}+\Delta x_{i}\right)=h\left(x_{i}{}^{0}\right)+J_{p}\Delta x_{i}$$
where *J_p_* is the Jacobian matrix with re-projection error, $\Delta x_{i}=x_{i}-x_{i}{}^{0}$, and $x_{i}{}^{0}$ is the initial pose.

Therefore, the error function can be expressed in the following form:(29)$$\{\begin{array}{c} e^{2}={\left\|h\left(x_{i}{}^{0}\right)+J_{p}\Delta x_{i}-\mu_{i}\right\|}^{2}={\left\|J_{p}\Delta x_{i}-b\right\|}^{2}\\ b=\mu_{i}-h\left(x_{i}{}^{0}\right) \end{array}$$

Then the pose error is solved. *S_ij_*, *S_i_*, and *S_j_* measurements can be obtained by a front-end visual odometry. The pose error function is as follows:(30)$$e_{ij}=\ln{\left(S_{ij}{}^{-1}S_{i}{}^{-1}S_{j}\right)}^{\vee }=\ln{\left(\exp\left({\left(-\varsigma_{ij}\right)}^{\wedge }\right)\exp\left({\left(-\varsigma_{i}\right)}^{\wedge }\right)\exp\left({\left(\varsigma_{j}\right)}^{\wedge }\right)\right)}^{\vee }$$

Next, derivation is made for $\varsigma_{i}$ and $\varsigma_{j}$, respectively. According to the derivation method of Lie algebra, $\varsigma_{i}$ and $\varsigma_{j}$ left-multiply by perturbations $\delta \varsigma_{i}$ and $\delta \varsigma_{j}$, respectively, so the error function becomes as follows:(31)$${\hat{e}}_{ij}=\ln{\left(S_{ij}{}^{-1}S_{i}{}^{-1}\exp\left({\left(-\delta \varsigma_{i}\right)}^{\wedge }\right)\exp\left({\left(\delta \varsigma_{j}\right)}^{\wedge }\right)S_{j}\right)}^{\vee }$$

By using the adjoint matrix property of similar transformation, the perturbation term can be moved to the rightmost side of the function. As a result, the Jacobian matrix in the form of right multiplication can be derived by using the Baker Campbell Hausdorff (BCH) formula. The error function is as follows:(32)$$\begin{array}{ll} {\hat{e}}_{ij} & =\ln{\left(S_{ij}{}^{-1}S_{i}{}^{-1}S_{j}\exp\left({\left(-Ad_{s}\left(S_{j}{}^{-1}\right)\delta \varsigma_{i}\right)}^{\wedge }\right)\exp\left({\left(Ad_{s}\left(S_{j}{}^{-1}\right)\delta \varsigma_{j}\right)}^{\wedge }\right)\right)}^{\vee }\\ & \approx \ln{\left(S_{ij}{}^{-1}S_{i}{}^{-1}S_{j}\left[I-{\left(Ad_{s}\left(S_{j}{}^{-1}\right)\delta \varsigma_{i}\right)}^{\wedge }+{\left(Ad_{s}\left(S_{j}{}^{-1}\right)\delta \varsigma_{j}\right)}^{\wedge }\right]\right)}^{\vee }\approx e_{{}_{ij}}^{0}+\frac{\partial e_{ij}}{\partial \delta \varsigma_{i}}\delta \varsigma_{i}+\frac{\partial e_{ij}}{\partial \delta \varsigma_{j}}\delta \varsigma_{j} \end{array}$$
where $e_{{}_{ij}}^{0}$ is the initial observation error value. The Jacobian matrix for $\varsigma_{i}$ and $\varsigma_{j}$ is:(33)$$\{\begin{array}{c} \frac{\partial e_{ij}}{\partial \delta \varsigma_{i}}=-J_{r}{\left(e_{ij}\right)}^{-1}Ad\left(S_{j}{}^{-1}\right)\\ \frac{\partial e_{ij}}{\partial \delta \varsigma_{j}}=J_{r}{\left(e_{ij}\right)}^{-1}Ad\left(S_{j}{}^{-1}\right) \end{array}$$
where:(34)$$J_{r}{\left(e_{ij}\right)}^{-1}=\left(I+\frac{1}{2}\left[\begin{array}{ccc} \omega^{\wedge }+\nu I & \tau^{\wedge } & -\tau \\ 0 & \omega^{\wedge } & 0\\ 0 & 0 & 0 \end{array}\right]\right),Ad\left(S_{j}{}^{-1}\right)=\left[\begin{array}{ccc} sR & t^{\wedge }R & -t\\ 0 & R & 0\\ 0 & 0 & 1 \end{array}\right]$$

In the incremental pose map optimization method based on a factor graph, the usual processing method is to fix the optimized pose, and only optimize the pose of the newly added points. In Equation (32), $\delta \varsigma_{i}=0$ is usually used, so the error function can be simplified as follows:(35)$${\hat{e}}_{ij}\approx e_{{}_{ij}}^{0}+\frac{\partial e_{ij}}{\partial \delta \varsigma_{j}}\delta \varsigma_{j}=e_{{}_{ij}}^{0}+J_{r}{\left(e_{ij}\right)}^{-1}Ad\left(S_{j}{}^{-1}\right)\delta \varsigma_{j}=e_{{}_{ij}}^{0}+J_{e}\delta \varsigma_{j}$$
where the Jacobian matrix of pose error is $J_{e}=J_{r}{\left(e_{ij}\right)}^{-1}Ad\left(S_{j}{}^{-1}\right)$.

Therefore, the pose error can be written as follows:(36)$$e_{{}_{ij}}^{2}=\frac{1}{2}{\left(e_{{}_{ij}}^{0}+J_{e}\delta \varsigma_{j}\right)}^{2}$$

Taking the first three rows of the Jacobian matrix of the pose error, *e_ij_*, we can get a 3 × 7 Jacobian matrix, ${J^{\prime }}_{e}$. In order to ensure the calculation of the minimization equation, the Jacobian matrix of the re-projection error, *e*, should be expanded to a 3 × 7 Jacobian matrix, ${J^{\prime }}_{p}$, and a one-dimensional scale variable shall be added to the variable ξ to become a seven-dimensional variable, $\varsigma_{j}$, and b is also extended to a seven-dimensional vector, $b^{\prime }$. Therefore, the re-projection error can be written as:(37)$$e^{2}={\left\|J_{p}\Delta x_{i}-b\right\|}^{2}={\left\|{J^{\prime }}_{p}\delta \varsigma_{j}-b^{\prime }\right\|}^{2}$$

The non-linear least squares formula using the Huber kernel function is changed to:(38)$$\begin{array}{l} X_{\max}=\underset{X}{\arg\min}\left(\sum_{i}H\left(h\left(x_{i}\right)-\mu_{i}\right)+\sum_{i,j}H\left(\ln\left(S_{ij}{}^{-1}S_{i}{}^{-1}S_{j}\right)\right)\right),\\ =\{\begin{array}{c} \frac{1}{2}{\left\|{J^{\prime }}_{p}\delta \varsigma_{j}-b^{\prime }\right\|}^{2}+\frac{1}{2}{\left(e_{{}_{ij}}^{0}+J_{e}\delta \varsigma_{j}\right)}^{2},\left|h\left(x_{i}\right)-\mu_{i}\right|\leq \beta_{1},and,\ln\left(S_{ij}{}^{-1}S_{i}{}^{-1}S_{j}\right)\leq \beta_{2},\\ \frac{1}{2}{\left\|{J^{\prime }}_{p}\delta \varsigma_{j}-b^{\prime }\right\|}^{2}+\beta_{2}\left(\left(e_{{}_{ij}}^{0}+J_{e}\delta \varsigma_{j}\right)-\frac{1}{2}\beta_{2}\right),\left|h\left(x_{i}\right)-\mu_{i}\right|\leq \beta_{1},and,other,\\ \beta_{1}\left(\left\|{J^{\prime }}_{p}\delta \varsigma_{j}-b^{\prime }\right\|-\frac{1}{2}\beta_{1}\right)+\frac{1}{2}{\left(e_{{}_{ij}}^{0}+J_{e}\delta \varsigma_{j}\right)}^{2},\ln\left(S_{ij}{}^{-1}S_{i}{}^{-1}S_{j}\right)\leq \beta_{2},and,other. \end{array} \end{array}$$

In the above part, the problem of optimizing the pose by re-projection and similarity transformation is transformed into the problem of least squares optimization using the process in which the camera pose and scale scaling factor are optimized by using the incremental pose map based on a factor graph. The selection of reference frames is significant for the optimization. In this paper, a simple method was proposed for the selection of reference frames. Firstly, the current reference frame is used as the reference frame to match the feature points of the image of its next key frame, and the number of matching feature points is counted and is taken as the reference matching number, m_base_, of the current reference frame. Then, the remaining sequence images are respectively matched with the reference frame by feature points, and the number m_other_ is counted. The number of m_other_ is compared with the number of references matching m_other_. If m_other_ ≥ 0.2*m_base_, the original reference frame remains unchanged. If m_other_ < 0.2*m_base_, the reference frame needs to be changed and the current key frame is taken as a new reference frame. The above operations are conducted for all the key frames of the remaining sequence images, which can automatically complete the updated iteration of the reference frame and improve the efficiency of the whole optimization process.

In the whole process of visual SLAM, global optimization plays a very important role, because it can reduce the pose error and scale drift error of the whole system in visual SLAM, improve the measurement accuracy, and ensure more accurate navigation and positioning. The conditions of global optimization are as follows: 1. When the closed loop appears in the system, the closed-loop characteristic is needed to optimize the whole system; and 2. when the reference frame passes through 50 frames and there is no closed loop in it, the whole information is needed to carry out a global optimization.

Generally, the global optimization method is to use the equation information of all incremental pose map optimization and calculate this by taking all pose Lie algebras, $\varsigma_{i}$, as unknown variables, so as to obtain the optimal solution of all poses. This result can be used as the initial value of the next optimization.

## 5. Experiments

The data set used in the indoor and outdoor environment experiments in this paper mainly comes from the open data set on the Internet, including the TUM data set, KITTI data set, and EuRoC MAV data set [26,27]. The method provided in this paper was compared with the widely used OBR_SLAM2 method. In the closed-loop detection experiment, in order to ensure the sufficiency of the experiment, a USB camera was used to collect image data, and the real-time image was used for the experiment.

### 5.1. Indoor Environmental Experiments

The indoor environment experiment mainly adopts the TUM data set and EuRoC MAV data set. The experimental results are shown in the following figure.

In Figure 5, the black line is the track curve drawn according to the true value provided by the data set, the blue line is the track drawn according to the actual measured key frame, and the red line represents the distance error between the measured value and the corresponding true value. According to the error values obtained in Figure 5, the experimental error comparison results in Table 1 can be obtained.

The results of Table 1 show that the incremental pose SLAM presented in this paper outperforms the experimental results of ORB-SLAM2. The root mean square error, mean error, median error, standard deviation of error, minimum error, and maximum error are all reduced, therefore, the positioning accuracy is further improved. The main reason is that the incremental pose optimization based on similarity transformation not only considers the optimization of the camera’s own pose but also takes into account the optimization of the relative pose based on the scale factor, which greatly improves the estimation accuracy of the camera pose. In this paper, inverse depth estimation based on a probability graph not only improves the accuracy of depth estimation but also improves its robustness.

The average tracking time in the experiment is relatively stable, as it benefits from the stable light intensity of the indoor environment. Additionally, the combination of the sparse direct method and the feature point method combines the advantages of the direct method and the feature point method, which improves the robustness of the tracking. Therefore, in the course of the experiment, there was no tracking failure.

The experimental comparison shows that the accuracy of using the TUM data set is better than that using the EuRoC data set. The main reason is that most of the TUM indoor data sets were obtained by hand-held cameras with slow speeds, which are relatively stable, and easy for tracking and depth estimation, so the error is relatively small. However, the data of the EuRoC data set was collected by Unmanned Air Vehicle (UAV) with greater volatility, which is susceptible to interference, and the image quality is relatively poor, affecting the estimation of depth, so the error is relatively large.

### 5.2. Outdoor Environmental Experiments

The KITTI data set was used for the outdoor experiment. It collects real image data of urban, rural, and highway environments by a car-mounted camera. It provides a large number of outdoor environment materials, which can evaluate the performance of computer vision in a vehicle environment. In this paper, sequences 08 data sets were used to carry out experiments. In the experiment, we used the GPS data as the benchmark to compare the experimental results of traditional ORB-SLAM2 and incremental pose SLAM. In Figure 6, the dotted line is the track drawn according to the GPS data, and the solid line is the track drawn according to the key frame in the experiment. The comparative results of outdoor experiments based on the experimental results are shown in Table 2.

The experimental results show that the trajectory of incremental pose SLAM is relatively smooth. Compared with GPS data, the experimental error is relatively small, and the scale drift phenomenon is relatively small. However, the relative error of traditional ORB-SLAM2 is relatively large, and the scale drift phenomenon is relatively serious. In this paper, an incremental pose map optimization SLAM based on similarity transformation is proposed to mitigate the influence of scale drift on the experimental results.

Generally speaking, the trajectories obtained by experiments show that in urban and rural roads, the trajectories obtained are relatively accurate and the positioning accuracy is relatively high because of the relatively slow speed and the number of feature points that can be extracted, which is convenient for camera positioning, depth estimation, and tracking. On the freeway, because of the fast speed, it may appear that the speed of computer image processing is slower than that of the vehicle, resulting in tracking failure, and fewer feature points can be extracted on the freeway, and its stability is relatively poor, thus reducing the positioning accuracy.

An important characteristic of the outdoor environment relative to the indoor environment is that the outdoor environment is susceptible to the influence of light. In the same environment, the feature extraction will change obviously in the light direction and backlight, which increases the difficulty of feature matching. By comparing the results, we can see that the sparse direct method based on histogram equalization proposed in this paper can effectively reduce the influence of light on feature points, ensure the assumption of photometric invariance, and improve the robustness of feature extraction. Therefore, the tracking effect of incremental pose SLAM is more stable and can be applied to various experimental environments. When there are loops in the motion of the vehicle camera, this method can extract features effectively, and use the bag-of-words model to complete image matching, so as to optimize the whole situation to eliminate errors and improve the accuracy of global positioning.

When there are fewer feature points or moving too fast, the camera easily loses its target. The repositioning method based on the bag-of-words adopted in this paper can effectively complete the camera relocation. The combination of the sparse direct method and the feature point method can effectively process images with relatively few feature points and accurately estimate the camera pose.

### 5.3. Closed-Loop Detection Experiment of A Hand-Held Camera

In this paper, a 720P USB camera was used for experiments. In the experiment, the IMU was fixed on the camera, and the hand-held camera was moved with the computer to get real-time images, and the data of the IMU were used as benchmark data for comparison, and the results of the closed loop detection were judged according to the comparison results. The experimental results are shown in the following figure.

It can be seen from Figure 7 that SLAM without closed-loop detection is prone to static error, especially when the handheld camera turns, its error increases relatively, and with the increase of the running route and time, its error will become larger and larger. When the camera passes through the same point, its cumulative error will be close to 4 m. However, when closed-loop detection is added to monocular vision SLAM, the scale drift and cumulative error of the monocular camera can be effectively reduced, and the positioning accuracy can be improved, and its RMSE can be reduced to 0.62 m. In this paper, closed-loop detection based on bag-of-words was mainly used. The bag-of-words uses a dictionary to represent the features of each image, and image matching can be done by searching the dictionary of each image, thus eliminating the cumulative errors caused by the camera in motion. In the process of dictionary generation, the K-means clustering method based on mean initialization can effectively prevent cluster centers from falling into the local optimum and improve the accuracy of image feature classification.

In order to test the performance of the closed-loop detection based on mean initialization proposed in this paper, many experiments were carried out, and the recall precision experimental data were obtained. In this paper, we used the data sets of TUM, KITTI, and EuRoC to experiment. We compared the closed-loop detection based on ORB features, the closed-loop detection based on the line band descriptor (LBD), and the closed-loop detection method proposed in this paper [36,37,38]. The experimental results are shown in Figure 8.

It can be seen from the experimental results that in the data sets of TUM, KITTI, and EuRoC, the experiments are carried out respectively, but the final experimental results are the same. The measurement accuracy is: Proposed > LBD > ORB. The method proposed in this paper improves the accuracy of closed-loop detection, which can effectively reduce the experimental error and improve the positioning accuracy of visual slam.

Through the experiments of hand-held cameras, we can also see that one of the major drawbacks of monocular SLAM is the uncertainty of scale. If there is no closed-loop detection, after the camera moves a circle, the starting point and the end point have a large offset. In order to improve the accuracy of closed-loop detection, the inverse depth estimation method based on a probability graph is used in front-end processing to improve the accuracy of initial depth estimation; in back-end optimization, the incremental pose optimization method based on similarity transformation is used to improve the accuracy of camera pose estimation. When the closed loop is detected in the camera motion process, global optimization will be carried out at the same time to further improve the accuracy of the scale factor and camera pose, so that the cumulative error of the whole system can be eliminated in a larger time scale.

The effectiveness and robustness of the proposed method were fully verified by indoor and outdoor environmental experiments using open data sets. Through the closed-loop detection experiment of the hand-held camera, the necessity of closed-loop detection in monocular vision SLAM was fully verified. Closed-loop detection reduced the overall error and ensured the accuracy of the system.

## 6. Conclusions

Visual odometry (VO) is a preliminary estimation of the pose and depth of the camera in monocular SLAM. Usually, the quality of the initial values has a significant impact on the whole system, and with good initial values, satisfactory results can be produced through optimization. In this paper, the direct method and the feature point method were used for pose estimation. The feature points were selected through the sparsity. When the number of feature points was less than a certain threshold, the direct method was used; otherwise, the feature point method was adopted. In order to ensure the photometric invariance of the direct method, histogram equalization was proposed to process the image to reduce the influence of photometric changes on the results. In this paper, a mixed inverse depth estimation method based on a probability graph was used to estimate the camera depth, which not only improved the estimation accuracy but also improved the robustness.

In the closed-loop detection, the bag-of-words was used to process. A dictionary was built by extracting the features of each image, and feature matching was carried out by looking up the dictionary. Bag-of-words improved the efficiency and accuracy of image matching. In the dictionary generation stage, a K-means clustering method based on mean initialization was proposed, which ensured the global optimum of the cluster center. In the whole process of SLAM, when the closed loop was detected by using the bag-of-words, the information of all nodes stored in the back-end optimization would be used for global optimization, so as to eliminate the cumulative error of the system, improve the accuracy of positioning and mapping, and ensure the consistency of the whole process.

In the back-end optimization, an incremental pose map optimization method based on similarity transformation was proposed. During the optimization process, the scale factor was taken into account by using similarity transformation to reduce the influence of scale drift on the results. The application of incremental pose map optimization contributed to the expansion of nodes. When a new node was added, only the pose between the new node and the associated node was optimized, which greatly reduced the computational complexity and improved the computational speed.

In conclusion, incremental pose map optimization SLAM for monocular vision based on similarity transformation can reduce the scale drift of a monocular camera, reduce the computational complexity of optimization, and improve the positioning accuracy, which has strong robustness.

## Figures and Tables

**Figure 1 sensors-19-04945-f001:**
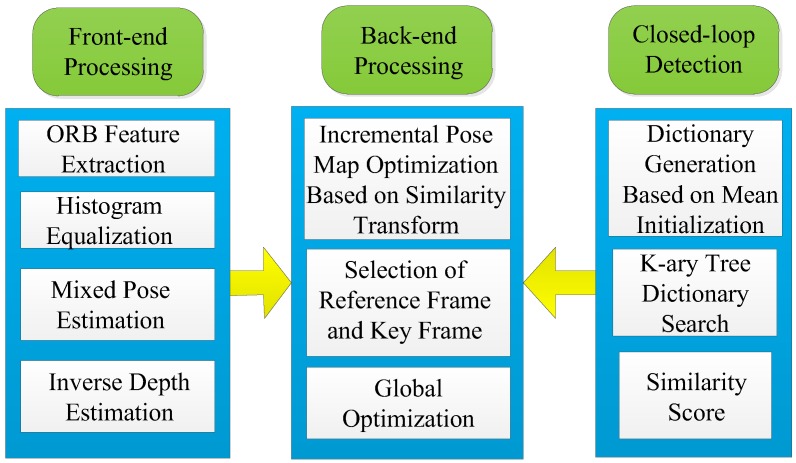
System schematic diagram of incremental pose map optimization for monocular vision SLAM based on similarity transformation.

**Figure 2 sensors-19-04945-f002:**
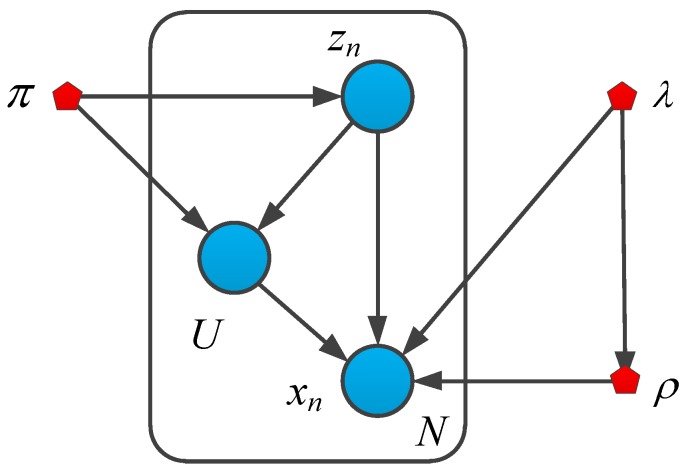
Probability diagram of Gauss-uniform mixed probability distribution model.

**Figure 3 sensors-19-04945-f003:**
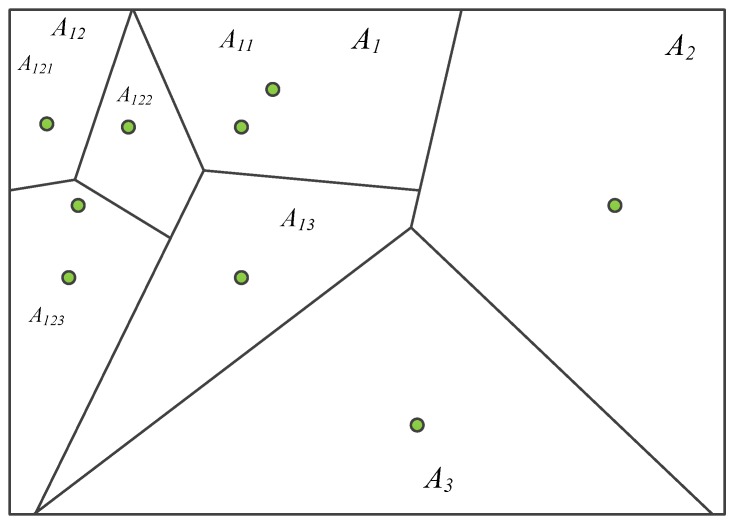
A schematic diagram of the generation process of a K-ary tree dictionary using the Thiessen polygon.

**Figure 4 sensors-19-04945-f004:**
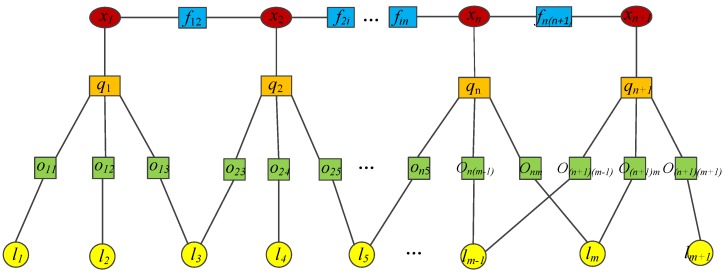
Schematic diagram of the incremental pose map optimization method.

**Figure 5 sensors-19-04945-f005:**
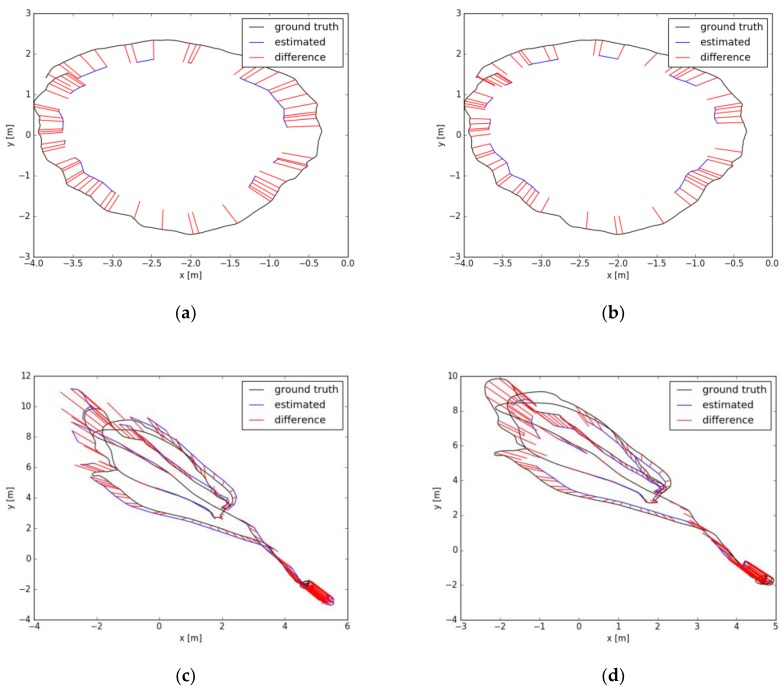
Comparison of TUM and EuRoC data sets before and after optimization. (**a**) Before TUM data set optimization; (**b**) After TUM data set optimization; (**c**) Before EuRoC data set optimization; (**d**) After EuRoC dataset optimization.

**Figure 6 sensors-19-04945-f006:**
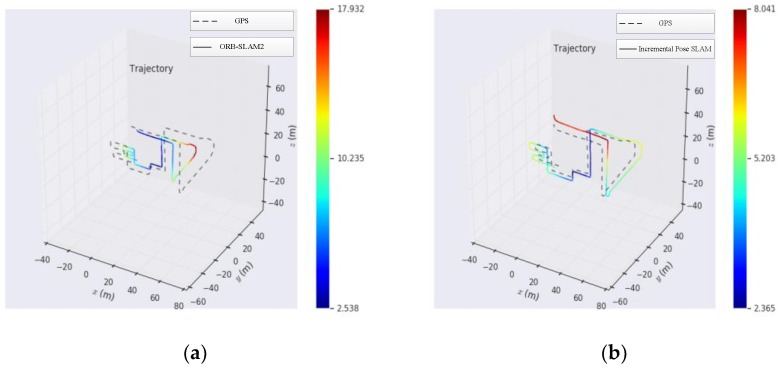
Experimental trajectory image of the KITTI dataset. (**a**) Comparison of GPS and ORB_SLAM2; (**b**) Comparison of GPS and incremental pose SLAM.

**Figure 7 sensors-19-04945-f007:**
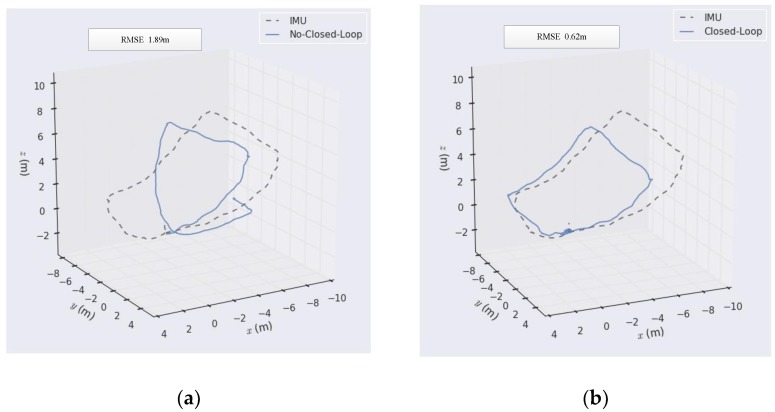
The experiment of the hand-held camera with or without closed-loop detection. (**a**) Comparison of the IMU and no-closed loop; (**b**) Comparison of IMU and closed-loop.

**Figure 8 sensors-19-04945-f008:**
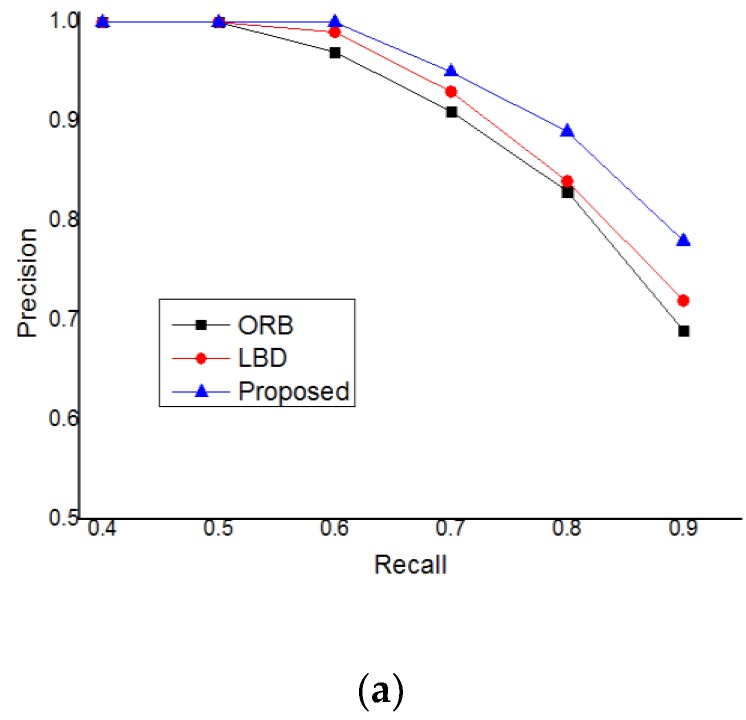
Precision-recall curves achieved by ORB, LBD, and proposed method. (**a**) Based on the data set of TUM; (**b**) Based on the data set of KITTI; (**c**) Based on the data set of EuRoC.

**Table 1 sensors-19-04945-t001:** Comparison of indoor experimental errors.

	TUM Data Set		EuRoC Data Set	
	ORB-SLAM2	Incremental Pose SLAM	ORB-SLAM2	Incremental Pose SLAM
Root mean square error (RMSE)	0.41 m	0.34 m	0.86 m	0.75 m
Mean error	0.40 m	0.34 m	0.76 m	0.66 m
Median error	0.41 m	0.35 m	0.81 m	0.69 m
Standard deviation of error	0.05 m	0.04 m	0.40 m	0.36 m
Minimum error	0.31 m	0.26 m	0.06 m	0.06 m
Maximum error	0.51 m	0.44 m	1.50 m	1.36 m
Average tracking time (s)	0.037 s	0.038 s	0.042 s	0.043 s

**Table 2 sensors-19-04945-t002:** Comparison of outdoor experimental errors.

	ORB-SLAM2	Incremental Pose SLAM
Root mean square error (RMSE)	10.52 m	6.25 m
Mean error	10.21 m	5.98 m
Median error	10.11 m	6.02 m
Standard deviation of error	4.68 m	2.01 m
Minimum error	3.89 m	2.13 m
Maximum error	18.98 m	10.63 m

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
