# Peer review of "Incremental Pose Map Optimization for Monocular Vision SLAM Based on Similarity Transformation"

_sensors, 2019, doi:10.3390/s19224945_

Round 1

Reviewer 1 Report

It is good to present precision and recall table for evaluating the performance of loop detection. However, we want to point out two problems and one recommendation. First problem is that authors do not inform what traditional closed-loop detection is. There are lots of loop detection methods to compare with, but the authors only use one of them with no reference. Use the several state-of-the-art loop detection methods given in the previous review and compare with yours. Second problem is that authors do not inform the datasets used for evaluating precision-recall. The information of datasets can be used to judge reliability of your experiment performance. Comment: when presenting precision-recall data, it is more visible on a plot than a table, so use a plot form. 

Author Response

It is good to present precision and recall table for evaluating the performance of loop detection. However, we want to point out two problems and one recommendation.

Point 1: First problem is that authors do not inform what traditional closed-loop detection is. There are lots of loop detection methods to compare with, but the authors only use one of them with no reference. Use the several state-of-the-art loop detection methods given in the previous review and compare with yours.

Response 1: Thank you for your comments.

I'm very sorry for the trouble caused by some unclear statements in this article. The traditional method mentioned in this paper mainly refers to the closed-loop detection method based on ORB features. In this paper, a new method, Line Band Descriptor (LBD), is added by referring to several latest literatures. Finally, ORB and LBD are compared with the methods proposed in this paper.

Thanks again for your comments.

Point 2: Second problem is that authors do not inform the datasets used for evaluating precision-recall. The information of datasets can be used to judge reliability of your experiment performance.

Response 2: Thank you for your comments.

I'm very sorry for our negligence in expression. In this paper, the main data set used in closed-loop detection is the TUM data set. This part has been re described in the paper, which increases the reliability and credibility of the paper. Finally, thank you very much for your suggestions, which have increased the persuasiveness of our paper.

Point 3: when presenting precision-recall data, it is more visible on a plot than a table, so use a plot form.

Response 3: Thank you for your comments.

  According to your suggestion, we have transformed tables into graphs to make the results more intuitive. Finally, thank you very much for giving us valuable suggestions for revision for three times in a row. You didn't give up our paper for my misunderstanding and some problems in the presentation of the paper. Every time I revise your opinions, I will study hard to make up for the loopholes and shortcomings of my knowledge. In the constant revision, I also gain knowledge and improve my ability.

  Thank you very much. Best wish to you.

Reviewer 2 Report

The paper concerns the problem of Simultaneous Localization and Mapping (SLAM). The authors addressed the problem of visual SLAM based on the monocular camera. They proposed an algorithm for the optimization of the incremental pose map. The algorithm is based on similarity transformation.

The paper is properly structured having a fair literature review on the problem in question and then a theoretical description of consecutive aspects of the approach. The performance of the approach was presented by the experiments on freely available public data sets. Although the results are not outstanding, they are well documented and the approach provides new knowledge in the domain.

Taking into account the aspect of the clearly presented and described the approach and fair reporting of the results from the experiments, I recommend the paper for the publication.

Author Response

Thank you very much for your suggestions and affirmation of our article. Best wish to you.

Reviewer 3 Report

Authors present an interesting approach to solve the problem of SLAM with an only cam. Despite this is a very exploited topic, there are several contributions which I consider interesting, such as depth estimation, cumulative error consideration an inverse parametrization, as an overall fusion of several well acknowledged methods.

Nonetheless, I feel there some aspects which should be improved:

Please reinforce the validity of the introduction with more up-to-date cites. They are valuable from the very beginning paragraphs (specifically to RGB-D, in order to compare and justify its benefits). There is an endless list of valid approaches in the MDPI database: Electronics, Remote Sensing, App Sciences, etc. I feel section 2 and 4 are difficult to follow for reader. I suggest equations are alleviated, and maybe more diagrams included to aid the comprehension. Figs 5-6-7 should add some more detail in the caption about the results shown. Info about legend, variables, etc. Higher resolution/size would be also apreciated.

Two main concers

The number of experiments. A wider/larger environment might be included for robustness checking. A better highlight of contributions should be clearly stated in the intro and also in the conclusion, with clear statement to the novelty in contrast to well-accepted approaches such as those from A.J Davison and Mur-Artal, amongs others.

Author Response

Please see the attachment for comments.

Reviewer 4 Report

The paper present an interesting study, which is aligned to Sensors Journal.

I propose some modification to the paper be published.

In the abstract, it is not clear the contribution of the work. It is not clear if the contribution is original. Which is the contribution regarding the literature? I suggest to present the literature gap and the novel contribution of the paper. I suggest to use “The novel contribution..”

Which open dataset was used?

Please, include some results in the abstract.

Please, consider the possibility of including words different from the title in the keywords.

Lines 31-32: Please, consider the possibility of change that this kind of study has also attracted the attention of many scholars. Your work can provide support to several interesting applications.

Lines 32-47: I suggest to include references to support the statements.

Lines 56, 59 and other: Verify the possibility of removing the use of the term “Document”.

In the introduction, it is also not clear the contribution of the work. Some related works were presented, but it is not clear the literature gap and the novel contribution of the paper. Is novel the used approach combining several techniques?

I suggest to include figure 1 in the method section.

Lines 165-171: Please, verify Sensors requirements regarding section calls. Section II or 2?

Sections 2, 3 and 4, should be included in a Method section with Figure 1.

I suggest to include the references for each used dataset.

Lines 543-544: It is not clear the phrase. Did you use USB camera to acquire images?

Lines 550-551: Please, review the phrase. You did not use the values obtained in Fig. 5. Fig. 5 only represents the trajectory.

Please, verify the possibility of representing one of the trajectories (blue or black) using dashed lines. It will more easy to differentiate them.

What are the results in Table 1? Is difference in distance?

Line 582: What is “sequence 08 data set”?

The RMSE is higher than 5 m. Is it really an accurate result? Is it a small error? Please, verify the use of “accurate” and “small” in the text. For 3D mapping, 5 m is not accurate.

Author Response

Please see the attachment for comments.

Round 2

Reviewer 1 Report

Loop detection experiment seems to be good, but there is still one minor points to revise. Authors should include reference of the conventional datasets such as TUM, KITTI, EuRoC datasets when these datasets are used for performance evaluation of the proposed method.

Author Response

Point 1: Loop detection experiment seems to be good, but there is still one minor points to revise. Authors should include reference of the conventional datasets such as TUM, KITTI, EuRoC datasets when these datasets are used for performance evaluation of the proposed method.

Response 1: Thank you for your comments.

Due to my lack of comprehensive thinking, there are many loopholes in the paper, which brings you a lot of trouble, so I am very sorry. In this paper, the experimental part of KITTI data set and EuRoC data set is added, and multiple data sets are used to verify the advantages of the proposed method. Combined with the results of comparison with the methods based on ORB and LBD, the persuasiveness of this method can be increased from different data sets and methods. Thank you very much for your valuable comments on the revision of my paper. Through the continuous revision of the paper, my ability is gradually improved and my thinking is more comprehensive.

Thank you very much. Best wish to you.

This manuscript is a resubmission of an earlier submission. The following is a list of the peer review reports and author responses from that submission.

Round 1

Reviewer 1 Report

The paper proposes an incremental pose map optimization for monocular vision SLAM based on similarity transformation, which can effectively solve the scale drift problem of SLAM for monocular vision and eliminate the cumulative error by global optimization. In addition, the combination of sparse direct method based on histogram equalization and feature-based method takes advantage of both. This is an innovative work and has achieved good experimental results. However, there are still some deficiencies in the content, experiment and expression of the article:
1. The introduction of the article only expounds the existing work from several aspects, lacking the targeted summary and own opinion of the existing work, such as contribution to this problem and shortcomings. The author should also explain the relationship between existing work and this article.
2. Is the loop closure detection method used in this paper the same as the method in ORB-SLAM? If yes, please reduce the length of the existing methods, if not, please highlight the differences. There is no need to introduce the word-bag method in a large space.
3. In the experimental part, is the data in Table 1 the average of all sequences in TUM and EUROC or the experimental results of partial sequences? In Figure 5, why is the estimated trajectory intermittent? Is the algorithm input all the frames in the dataset?
4. Lack of outdoor experimental error data, as in Tabel 1. In addition, it is more important that the article needs to do a comparative experiment to verify the role of each part of the proposed method. For example, the benefits of incremental optimization for estimation accuracy and the benefits of mixed features (direct method and features) for robustness.

This is a good work, but the organization and experimentation of the article needs to be greatly improved. In summary, I think that the paper is publishable after major revision.

Author Response

Dear,

    My answer is in the uploaded file.

Reviewer 2 Report

This paper presents an incremental pose map optimization for monocular vision SLAM based on similarity transformation. In introduction, related works are just listed through 3 paragraphs. It would be better to explicitly analyze and summarize the related works. Also the novelty of this paper is not shown properly. In section 2, the author proposed that direct method and feature point method are combined to make full use of the advantage of both and avoid disadvantage. Firstly, the author doesn’t discuss about the advantage and disadvantage of each method properly. Secondly, just flipping between two methods by checking number of features does not support author’s statement. In section 4, the authors use sim(3) for camera pose estimation. However, authors did not use sim(3) in section 2. It would be better to explain how authors used the different transformation matrix in front-end and back-end. In experiment, the author present only a table for the indoor environmental experiments. It would be better to present the experiment result about outdoor environment. Also, to evaluate the loop detection performance, the author need to use a recall-precision plot. The format and grammar should be rechecked. There are some minor mistakes. Some of mistakes are written below. In page 2: beam adjustment à bundle adjustment In page 3: BA adjustment à BA or Bundle Adjustment In Figure 1. : Maxed Pose Estimation à Mixed Pose Estimation In page 4: visual odometer à visual odometry

Author Response

(The authors gave the same response as above.)

Round 2

Reviewer 1 Report

The author has done enough work and the revision of the article is reasonable.

Author Response

Thank you for your question, which will enable us to think more deeply.

Reviewer 2 Report

Overall: Most questions are answered adequately. A remaining comment is given below.

Previous review comment 4:

In experiment, the author present only a table for the indoor environmental experiments. It would be better to present the experiment result about outdoor environment. Also, to evaluate the loop detection performance, the author need to use a recall-precision plot.

Your answer:

(1) In indoor experiments, TUM data sets and EuRoC data sets are used, which provide real pose at each moment and can be used for comparison, so Table 1 can be obtained. KITTI data set is used in outdoor environment experiment, which only provides image and time sequence, but does not provide real pose at each moment, so it is impossible to get data similar to Table 1.

(2) In this paper, in outdoor environment experiments, we qualitatively compare the incremental pose map optimization for monocular vision SLAM based on similarity transformation and traditional ORB-SLAM method. The validity of the proposed method can also be fully illustrated by the comparative experiment.

(3) We have reprocessed the data of the closed-loop detection experiment and plotted the two experimental results in the same picture, so the contrast effect is more obvious.

In the end, we would like to express our sincere thanks to you for providing us with valuable comments.

Review comments:

For the KITTI dataset, the authors can use GPS data as ground truth. In the ORB SLAM, they compared the result of the proposed method with the GPS data to evaluate.

As the authors proposed the loop detection method, it is essential to evaluate the proposed method. Precision-recall plot is mostly used to evaluate loop detection performance. So the authors should conduct the loop detection experiment and evaluate proposed loop detection method by precision-recall plot.

If this cannot be done in a timely manner, the manuscript should be withdrawn.

Author Response

Previous review comment 4:

In experiment, the author present only a table for the indoor environmental experiments. It would be better to present the experiment result about outdoor environment. Also, to evaluate the loop detection performance, the author need to use a recall-precision plot.

Response 4: Thank you for your comments.

Thank you very much for your suggestion. According to your suggestion, we reprocessed the experimental data and compared the experimental results with GPS data in the experimental process. The latest experimental results and tables are added to the paper. When I first revised my paper, I misunderstood what you meant. I thought you wanted me to compare closed-loop and no-closed-loop data together. I am very sorry for my misunderstanding. I used IMU data as a benchmark, and did the experiment again. The experimental results have been added to the paper.

Thank you for your question, which will enable us to think more deeply.

Round 3

Reviewer 2 Report

Previous review comment 4:

In experiment, the author present only a table for the indoor environmental experiments. It would be better to present the experiment result about outdoor environment. Also, to evaluate the loop detection performance, the author need to use a recall-precision plot.

Your answer (1):

(1) In indoor experiments, TUM data sets and EuRoC data sets are used, which provide real pose at each moment and can be used for comparison, so Table 1 can be obtained. KITTI data set is used in outdoor environment experiment, which only provides image and time sequence, but does not provide real pose at each moment, so it is impossible to get data similar to Table 1.

(2) In this paper, in outdoor environment experiments, we qualitatively compare the incremental pose map optimization for monocular vision SLAM based on similarity transformation and traditional ORB-SLAM method. The validity of the proposed method can also be fully illustrated by the comparative experiment.

(3) We have reprocessed the data of the closed-loop detection experiment and plotted the two experimental results in the same picture, so the contrast effect is more obvious.

In the end, we would like to express our sincere thanks to you for providing us with valuable comments.

Review comments (1):

For the KITTI dataset, the authors can use GPS data as ground truth. In the ORB SLAM, they compared the result of the proposed method with the GPS data to evaluate.

As the authors proposed the loop detection method, it is essential to evaluate the proposed method. Precision-recall plot is mostly used to evaluate loop detection performance. So the authors should conduct the loop detection experiment and evaluate proposed loop detection method by precision-recall plot.

If this cannot be done in a timely manner, the manuscript should be withdrawn.

Your answer (2):

Thank you for your comments.

Thank you very much for your suggestion. According to your suggestion, we reprocessed the experimental data and compared the experimental results with GPS data in the experimental process. The latest experimental results and tables are added to the paper. When I first revised my paper, I misunderstood what you meant. I thought you wanted me to compare closed-loop and no-closed-loop data together. I am very sorry for my misunderstanding. I used IMU data as a benchmark, and did the experiment again. The experimental results have been added to the paper.

Thank you for your question, which will enable us to think more deeply.

Review comments (2):

The authors seem to misunderstand our comment on the loop detection experiment. The papers below present loop detection methods and precision-recall plots to evaluate their methods. As authors propose a loop detection method, precision-recall plot should be used to evaluate the proposed method.

“FAB-MAP: Probabilistic Localization and Mapping in the Space of Appearance,” International Journal of Robotics Research, vol. 27, no 6, pp. 647-665, 2008. “BRIEF-Gist – Closing the Loop by Simple Means,” International Conference on Intelligent Robots and Systems (ICRA 2011), San Francisco, USA, Sep. 25-30, 2011. “Bag of Binary Words for Fast Place Recognition in Image Sequences,” IEEE Transaction on Robotics, vol. 28, no. 5, pp. 1188-1197, 2012. “On the Performance of ConvNet Features for Place Recognition,” International Conference on Intelligent Robots and Systems (IROS 2015), Hamburg, Germany, Sep. 28 – Oct. 2, 2015. “A Monocular Vision Sensor-Based Efficient SLAM Method for Indoor Service Robots,” IEEE Transactions on Industrial Electronics, vol. 66, no. 1, pp. 318-328, 2019.

If this cannot be done in a timely manner, the manuscript should be withdrawn.